# The Experimental Proteome of *Leishmania infantum* Promastigote and Its Usefulness for Improving Gene Annotations

**DOI:** 10.3390/genes11091036

**Published:** 2020-09-02

**Authors:** África Sanchiz, Esperanza Morato, Alberto Rastrojo, Esther Camacho, Sandra González-de la Fuente, Anabel Marina, Begoña Aguado, Jose M. Requena

**Affiliations:** Centro de Biología Molecular “Severo Ochoa” (CBMSO, CSIC-UAM) Campus de Excelencia Internacional (CEI) UAM+CSIC, Universidad Autónoma de Madrid, 28049 Madrid, Spain; africa.sanchiz@gmail.com (Á.S.); emorato@cbm.csic.es (E.M.); arastrojo@cbm.csic.es (A.R.); ecamacho@cbm.csic.es (E.C.); sandra.g@cbm.csic.es (S.G.-d.l.F.); amarina@cbm.csic.es (A.M.); baguado@cbm.csic.es (B.A.)

**Keywords:** *Leishmania infantum*, proteome, post-translational modifications (PTMs), proteogenomics, mass spectrometry

## Abstract

*Leishmania infantum* causes visceral leishmaniasis (kala-azar), the most severe form of leishmaniasis, which is lethal if untreated. A few years ago, the re-sequencing and de novo assembling of the *L. infantum* (JPCM5 strain) genome was accomplished, and now we aimed to describe and characterize the experimental proteome of this species. In this work, we performed a proteomic analysis from axenic cultured promastigotes and carried out a detailed comparison with other *Leishmania* experimental proteomes published to date. We identified 2352 proteins based on a search of mass spectrometry data against a database built from the six-frame translated genome sequence of *L. infantum*. We detected many proteins belonging to organelles such as glycosomes, mitochondria, or flagellum, as well as many metabolic enzymes and many putative RNA binding proteins and molecular chaperones. Moreover, we listed some proteins presenting post-translational modifications, such as phosphorylations, acetylations, and methylations. On the other hand, the identification of peptides mapping to genomic regions previously annotated as non-coding allowed for the correction of annotations, leading to the N-terminal extension of protein sequences and the uncovering of eight novel protein-coding genes. The alliance of proteomics, genomics, and transcriptomics has resulted in a powerful combination for improving the annotation of the *L. infantum* reference genome.

## 1. Introduction

The genus *Leishmania* belongs to the order Trypanosomatida and includes protozoan parasites that are responsible for a complex of diseases named leishmaniasis, which is the second most common cause of mortality among tropical infectious diseases, after malaria [1]. Some species of *Leishmania* are human pathogens that cause different clinical manifestations as cutaneous (CL), mucocutaneous (MCL), or visceral (VL) leishmaniasis. Old World species *Leishmania infantum* and *Leishmania donovani* cause VL or kala-azar, which is often lethal if untreated, whereas *Leishmania major* causes CL and *Leishmania braziliensis* is associated with MCL. The VL-causative species are genetically almost identical, although they differ in geographic distribution: *L. donovani* is found in the Indian subcontinent and East Africa, while *L. infantum* is endemic in the countries around the Mediterranean basin, Latin America, and China [2]. Two stages, promastigote and amastigote, alternate in the *Leishmania* life cycle. Promastigotes are flagellated and motile forms develop extracellularly in the gut of their sand-fly vector. The infection of the mammal host takes place during the sand-fly blood meal; afterwards, parasites are phagocytized by macrophages, and amastigote forms develop inside these host cells.

Within the *Leishmania* genus, the *L. major* genome was the first to be sequenced [3], followed by the *L. infantum* and *L. braziliensis* ones [4]. During the last decade, the extraordinary progress in DNA sequencing methodologies has allowed for the drafting of the genomes for many other *Leishmania* species [5,6,7,8,9,10,11,12] and for the improvement of the assemblies of the first sequenced genomes [13,14,15].

The availability of a complete and well-annotated genome provides the ultimate resource for genome-wide scale approaches, such as transcriptome and proteome analyses [16]. In parallel to the advances in sequencing technologies, proteomics methodologies are achieving unprecedented levels of sensitivity, and novel MS-based experimental approaches have become the method of choice for the analysis of complex protein mixtures such as cells, tissues, and even whole organisms. In particular, several proteomic technologies are being used to study diverse aspects of *Leishmania* biology such as parasite development, virulence, and drug resistance [17]. Thus, proteomics approaches have been used to determine differential patterns of protein expression between the promastigote and amastigote stages in *Leishmania mexicana* [18], *L. infantum* [19], and *L. donovani* [20], among others. Other studies have been aimed to ascertain specific proteomes by means of organelle fractionation to obtain enriched fractions of mitochondria, flagella, or glycosomes [21,22]. The identification and mapping of protein post-translational modifications (PTMs) provide additional information about the activation of specific pathways in a given growing condition, thus improving the knowledge on protein interactions in complex networks.

Here, we present a wide and detailed proteome of the *L. infantum* JPCM5 strain, based on axenically grown promastigotes in the logarithmic growth phase. A careful comparative analysis with other published proteomes from different Old and New World *Leishmania* species has also been carried out. Additionally, the MS data allowed for the identification of PTMs (phosphorylation, acetylation, methylation, formylation, and glycosylation) at specific protein sites that might have regulatory functions. Furthermore, we showed how the integration of proteomics with genomic and transcriptomic data represents a powerful and complementary strategy for gene annotation, as demonstrated before in a plethora of species [23]. Hence, by applying this proteogenomic approach, it was possible to improve the annotations for several *L. infantum* genes, as well as the identification of eight novel genes.

## 2. Materials and Methods

### 2.1. Leishmania Infantum Culture and Protein Extraction

*L. infantum* JPCM5 strain parasites were grown at 26 °C in Roswell Park Memorial Institute (RPMI) 1640 medium supplemented with 15% of heat inactivated fetal calf serum (Biowest SAS, Nuaillé, France). Promastigote cultures were initiated at 1 × 10^6^ parasites/mL and harvested at the mid-logarithmic phase (1–2 × 10^7^ parasites/mL). Around 1–2 × 10^8^ parasites were collected and washed twice with phosphate-buffered saline (PBS); finally, parasites were suspended by pipetting in 300 µl of a RIPA (RadioImmunoPrecipitation Assay) lysis buffer (Thermo Fisher Scientific, Rockford, IL, USA) in the presence of EDTA-free Easy Pack Protease inhibitor (Roche, Diagnostics, Mannheim, Germany). After 6 cycles (30 s pulse/30 s pause) of sonication in a bath at 4 °C, samples were incubated for 90 min at 4 °C, and, afterwards, protein lysates were centrifuged at 14,000 g for 30 min. The supernatant was collected and used for proteomics analyses.

### 2.2. In-Gel and In-Solution Digestion of Samples by Trypsin and Chymotrypsin

For the in-gel digestion of proteins, samples were mixed with an equal volume of a 2× Laemmli buffer and loaded onto 1.2-cm wide wells of a conventional SDS-PAGE gel (0.75 mm-thick, 4% polyacrylamide stacking-gel, and 10% polyacrylamide resolving-gel). The electrophoresis was stopped as soon as the electrophoretic front entered 3 mm into the resolving gel, so the proteins became concentrated in the stacking/resolving gel interface. After Coomassie staining, the protein-containing gel was cut into small pieces (2 × 2 mm cubes) and placed into a microcentrifuge tube, as described elsewhere [24]. The gel pieces were destained in acetonitrile:water (ACN:H_2_O, 1:1), reduced and alkylated (disulfide bonds from cysteinyl residues were reduced with 10 mM dithiothreitol (DTT) for 1 h at 56 °C, and then thiol groups were alkylated with 10 mM iodoacetamide for 1 h at room temperature in darkness), and digested in situ with sequencing grade trypsin (Promega, Madison, WI) or chymotrypsin (Roche Diagnostics), as described by Shevchenko et al. [25], with minor modifications. The gel pieces were shrunk by removing all liquid using sufficient ACN. Acetonitrile was pipetted out, and the gel pieces were dried in a speedvac. The dried gel pieces were re-swollen in 100 mM Tris-HCl and 10 mM CaCl_2_ at pH 8 with 60 ng/µL trypsin or chymotrypsin at a 5:1 protein:enzyme (*w*/*w*) ratio. The tubes were kept on ice for 2 h and incubated at 37 °C (trypsin) or 25 °C (chymotrypsin) for 12 h. Digestion was stopped by the addition of 1% trifluoroacetic acid (TFA). Whole supernatants were dried down and then desalted onto OMIX C18 pipette tips (Agilent Technologies, Santa Clara, CA, USA) before the MS analysis.

Additionally, in-solution digestion was performed as described elsewhere [26]. After the denaturation of proteins with an 8 M urea, the protein sample was reduced and alkylated: disulfide bonds from cysteinyl residues were reduced with 10 mM DTT for 1 h at 37 °C, and then thiol groups were alkylated with 50 mM iodoacetamide for 1 h at room temperature in darkness. The sample was diluted to reduce urea concentration below 1.4 M and digested using sequencing-grade trypsin (Promega, Madison, WI, USA) or chymotrypsin (Roche Diagnostics) overnight at 37 °C (trypsin) or 25 °C (chymotrypsin) using a 1:20 (*w*/*w*) enzyme/protein ratio. Digestion was stopped by the addition of 1% TFA. Whole supernatants were dried down and then desalted onto OMIX C18 pipette tips (Agilent Technologies) before the MS analysis.

### 2.3. Reverse Phase-Liquid Chromatography Mass Spectrometry Analysis (RP-LC-MS/MS)

The digested protein samples (above) were resuspended in 10 µl of 0.1% formic acid and analyzed by RP-LC-MS/MS in an Easy-nLC II system coupled to an ion trap LTQ-Orbitrap-Velos-Pro hybrid mass spectrometer (Thermo Fisher Scientific). The peptides were concentrated (on-line) by reverse phase chromatography using a 0.1 × 20 mm C18 RP precolumn (Thermo Fisher Scientific) and then separated using a 0.075 × 250 mm C18 RP column (Thermo Fisher Scientific) operating at 0.3 μL/min. Peptides were eluted using a 180-min dual gradient. The gradient profile was set as follows: 5−25% solvent B for 135 min, 25−40% solvent B for 45 min, 40−100% solvent B for 2 min, and 100% solvent B for 18 min (Solvent A: 0.1% formic acid in water; solvent B: 0.1% formic acid and 80% acetonitrile in water). ElectroSpray ionization (ESI) was done using a nano-bore emitter stainless steel ID 30 μm (Proxeon) interface. The Orbitrap resolution was set at 30,000. Peptides were detected in survey scans from 400 to 1600 amu (1 μscan), followed by twenty data-dependent MS/MS scans (Top 20) using an isolation width of 2 u (in mass-to-charge ratio units), a normalized collision energy of 35%, and a dynamic exclusion that was applied during 60 s periods.

### 2.4. Data Analysis

Peptide identification from raw data was carried out using the PEAKS Studio X search engine (Bioinformatics Solutions Inc, Waterloo, ON, Canada). A custom Python script was used to create a database comprising all possible open reading frames (ORF) coding for protein sequences of ≥20 amino acids existing in any of the six-frames in the *L. infantum* JPCM5 strain genome sequence [13]. This database (named LINF-all-ORFs) consisted of 294,654 entries. In parallel, a fusion-database, created by merging the *L. infantum* protein sequences annotated in UniProt and the LINF-all-ORFs entries, was also used by the search engine. Finally, a search against a decoy database (decoy fusion-database) was also performed. The following constraints were used for the searches: tryptic or chymotryptic cleavage (semispecific), up to two missed cleavage sites, tolerances of 20 ppm for precursor ions and 0.6 Da for MS/MS fragment ions, and optional Met oxidation and Cys carbamidomethylation were allowed. The false discovery rates (FDRs) for peptide spectrum matches (PSMs) were limited to 0.01 or lower. Those proteins that were identified with at least two distinct peptides were considered for further analysis [27,28,29].

The LINF-all-ORFs entries with mapped peptides were compared with the annotated *L. infantum* proteins (UniProt database) using in-house Python scripts in order to identify both the misannotations and novel proteins. Additionally, Python scripts were used to ascribe post-translational modifications to particular protein entries.

Functional categories and enzymatic pathways using the DAVID program (Functional Annotation Tool, DAVID Bioinformatics Resources 6.8) and the KEGG Pathway (Kyoto Encyclopedia of Genes and Genomes) were used for the classification of the proteins identified by MS.

## 3. Results and Discussion

### 3.1. Protein Identification from the LC−MS/MS Peptide Spectra

The main objective of this study was to obtain the experimental proteome of the *L. infantum* promastigote stage with the additional aim of improving current genomic annotations. For this purpose, a recently published re-sequenced genome [13] was used. However, we did not restrict the search of peptide spectra on currently annotated protein-coding genes; instead, a database consisting of all possible polypeptides (equal or larger than 20 amino acids) was used (see Materials and Methods for further details). The workflow used for sample preparation and proteomics is shown in Figure 1. From the MS/MS data, it could be seen that only those associated with peptides longer than seven amino acids were considered for protein identification. Among the identified proteins, 2344 proteins matched with previously annotated proteins [13]. Moreover, eight novel proteins were uncovered, thus legitimating the search strategy. In addition, some ORFs had to be extended to accommodate the MS-identified peptides (see below for further details about these findings). Most of the proteins—70.5% (1659 out of 2352)—were identified by three or more unique peptides per protein, 14.5% (341) of the protein identifications were supported by two unique peptides, and only 15% (352) of the identifications were done by a single unique peptide. Currently, 3482 out of 8590 annotated proteins (around 40%) in the *L. infantum* genome have the status of hypothetical proteins; the MS spectra obtained in this work provided experimental evidence of the real existence for 456 of those hypothetical proteins.

The first comprehensive study aimed to characterize the *L. infantum* proteome was carried out by the Papadopoulou’s group [30]. Using two-dimensional (2D) gel electrophoresis, these authors visualized 2261 protein spots in promastigote samples and 2273 spots in amastigote ones. However, after MS analysis, only 168 protein spots, derived from 71 different genes, could be identified [30]. A better proteome resolution was attained after a fractionation step including digitonin extraction; hence, 153 *L. infantum* proteins were identified by MS analysis of selected spots [31]. The combination of two-dimensional liquid chromatography (2DLC), electrospray ionization mass spectrometry (2DLC-ESI-MS), and 2DLC-matrix-assisted laser desorption/ionization mass spectrometry (2DLC-MALDI-MS) allowed Leifso and co-workers to identify 91 *L. infantum* proteins [19]. An enrichment for basic proteins using the technique of free-flow electrophoresis prior to separation by 2D gel electrophoresis led to the identification of around 200 *L. infantum* proteins [32]. Alcolea and coworkers [33] identified 28 proteins in a proteomic study aimed to uncover differentially expressed proteins between the early-logarithmic and the stationary phases during the culturing of *L. infantum* promastigotes. In two different studies using MS analysis of the exoproteome derived from *L. infantum* promastigote cultures, a total of 102 [34] and 494 [35] proteins were identified. Therefore, our work provides the most complete, to date, experimentally evidenced proteome for *L. infantum*.

Outstanding studies on proteome identification have been performed in both *L. donovani* and *L. major*. In 2008, Rosenzweig and collaborators reported the identification of 1713 proteins in *L. donovani* [20]. A comparison between the proteins identified in our work (*L. infantum* JPCM5) and those identified in *L. donovani* showed that 1218 proteins were common (orthologs) in both studies (Figure 2). We failed to identify 207 proteins of those reported in *L. donovani*, whereas we found 1130 proteins that are absent from the *L. donovani* proteome reported by Rosenzweig et al. [20]

More recently, Pandey and coworkers reported the identification of 3386 different proteins in *L. donovani* promastigote and amastigote stages [37,38]. After comparing their data and the proteins identified in this study, 1650 of the proteins observed in *L. infantum* promastigotes were found to be present (their orthologues) in the *L. donovani* promastigote proteome. However, among the 613 proteins that Nirujogi et al. [37] reported to be exclusively expressed in *L. donovani* amastigotes, 126 proteins were also identified in our proteomics study, thus indicating that these proteins are also being expressed in the promastigotes stage, at least in *L. infantum* (see Appendix A). Most of them were annotated as hypothetical proteins or with unknown function, but there are also metabolic enzymes, translation machinery components (ribosomal proteins and eukaryotic initiation factors), and RNA binding proteins.

In 2014, Pawar et al. [36] reported a quite wide proteome of the *L. major* promastigote stage, in which 3613 proteins were identified. These authors followed a proteogenomic approach, as we did in this study, consisting of searching the mass spectra against a six-frame translated database generated from a complete genome sequence. An orthology-based comparison indicated that the *L. major* promastigote proteome and the *L. infantum* proteome of this study shared 1733 proteins (Figure 2). Moreover, considering the 1792 proteins identified in the *L. major* proteome, though not in our study, and the 615 proteins exclusively identified by us in the *L. infantum* proteome, the total number of identified proteins presumably expressed in the promastigote stage is 4140 (roughly half of the predicted proteins to be encoded in the *Leishmania* genome).

### 3.2. Representativeness of the Translational Machinery and RNA Binding Proteins in the L. infantum Experimental Proteome

Around 200 proteins from the *L. infantum* promastigote proteome were categorized as components of the translational machinery: 122 ribosomal proteins, 51 translation regulatory factors, and 24 tRNA synthases (Appendix A). As expected for a highly proliferative stage (the promastigotes were growing in the logarithmic phase when harvested for analysis), in which protein synthesis needs to be very active, all the ribosome components, tRNA-loading enzymes, and regulatory factors were abundant and easily detected by mass spectrometry. Nevertheless, in contrast, very few of the annotated mitoribosomal proteins [39] were identified in the *L. infantum* promastigote proteome. This observation may indicate that mitochondrial ribosomes are in relatively low amounts when promastigotes are grown in nutrient-rich culture media.

Proteins with RNA binding properties deserve special attention, since gene expression in *Leishmania* and related trypanosomatids is essentially being controlled at the post-transcriptional level [40,41]. In this scenario, RNA-binding proteins are key players in controlling RNA metabolism [42,43,44]. In the *L. infantum* promastigote proteome, a large number of known RNA binding proteins were detected (Appendix A). Apart from the mentioned ribosomal proteins and translation factors, 15 RNA helicases were detected, as well as many of the RNA-binding domain-containing proteins reported in a recent study aimed to the capture and identification of RNA-bound proteins in *L. donovani* [45]. The RNA-binding proteins of the Pumilio family (aka PUF proteins) are especially numerous (11 members) in *Leishmania* [46]. In this study, we identified 6 out of 11 PUF proteins that are being expressed in the promastigote stage of *L. infantum*; these are PUF 1, PUF 4, PUF 6, PUF 7, PUF 8, and PUF 10 (see Appendix A to see their gene IDs).

### 3.3. Metabolic Enzymes and Pathways

Going deeper, we performed an in-silico pathway reconstruction using the detected proteins in the *L. infantum* promastigote proteome. By using the KEGG database resource (http://www.genome.jp/kegg/) accessed via the DAVID package, a total of 578 (27.6%) of the detected proteins could be classified into pathways representing classical cellular processes. In particular, 236 proteins were identified as metabolic enzymes; 31% of these enzymes belong to glycolysis (Table 1 and Table 2), the tricarboxylic acid (TCA) cycle, and the pentose phosphate cycle (Appendix A), which are three metabolic pathways playing essential maintenance functions in the cell [47]. Remarkably, the complete set of 29 enzymes that make up the TCA cycle were identified in the promastigote proteome (Figure 3).

The glycosome is a trypanosomatid-specific, membrane-enclosed organelle that contains glycolytic enzymes, among others. Thus, glycolysis in *Leishmania* takes place in these organelles for the steps between glucose and 3-phosphoglycerate [48], as well as in the cytosol for those late steps leading to the formation of pyruvate [20]. The identified enzymes involved in these two stages are listed in Table 1 and Table 2. Among them, there are 32 enzymes belonging to the glycosomal/cytosolic glycolysis (and gluconeogenesis) pathway until the formation of pyruvate by pyruvate kinases (IDs LINF_350005400 and LINF_350005300). Some enzymes involved in the mitochondrial electron transport respiratory chain were detected (Appendix A). Similar findings were found by Rosenzweig and collaborators in the *L. donovani* promastigote proteome [20]. Several proteins of the electron transport chain are encoded by the kinetoplast DNA maxicircles [49] such as cytochrome oxidase subunits and NADH dehydrogenase, but they were not searched in this study.

An active energy metabolism requires enzymes to be involved in redox homeostasis. Several of these enzymes have been identified in the *L. infantum* proteome, and they are likely abundant, as judged by the large number of unique peptides that were mapped to them. The detected proteins were tryparedoxin (LINF_150019000, with 31 unique peptides), peroxidoxin (LINF_230005400, 31 peptides), cyclophilin (LINF_060006300, 20 peptides), iron superoxide dismutases (LINF_080007900 and LINF_320024000, with 14 and 18 peptides, respectively), and several elongation factors 1β (LINF_340014200 and LINF_340014000 with 16 peptides each and LINF_360020500 with 19 peptides).

### 3.4. Components of the Proteostasis Network

The proteasome is a complex of multi-subunit proteases, associated with protein degradation, but in protozoan parasites, it has been also involved in cell differentiation and replication processes [50]. In fact, proteasomal inhibitors have been described as promising therapeutic targets for leishmaniasis and trypanosomiasis [51,52]. According to the KEGG database, the complete compendium of proteasomal proteins were identified in this study (Appendix A).

Protein degradation and protein folding cooperate to maintain protein homeostasis or proteostasis [53]. Multiple and drastic environmental changes (pH variation, sudden temperature up-/down-shifts, and oxidative stress) occur along the *Leishmania* life cycle. Most often, these environmental insults promote protein unfolding and aggregation; to counteract these effects, cells possess specialized molecular chaperones (or heat shock proteins: HSPs) that serve as central integrators of protein homeostasis. Not surprisingly, *Leishmania* parasites possess a large number and variety of molecular chaperones [54]. In this study, we identified proteins belonging to the different HSP families: HSP100, HSP83/90, HSP70, HSP60, HSP40/DnaJ, and HSP20 (listed in Appendix A). Mitochondrion is a cellular organelle in which molecular chaperones are of particular relevance because they are involved in protein transport across membranes and protein refolding inside the mitochondria. Recently, the mitochondrial proteome was analyzed in *L. tropica* [22]. Taking advantage of that study, in Table 3, we list those HSPs identified in the *L. infantum* proteome that are potentially mitochondrial proteins.

### 3.5. Glycosomal Proteins Represent a Substantial Fraction of the Experimentally Detected Proteins in the L. infantum Promastigote

As mentioned above, glycosomes are specialized peroxisomes that contain key enzymes involved on energy metabolism and purine salvage [55]. Moreover, as occurs in peroxisomes, glycosomal proteins are targeted for import to and location in glycosomes by the presence of the peroxisomal targeting signals (PTSs) PTS1 and PTS2. Two essential proteins for targeting newly synthesized proteins, with a PTS2 import signal, into the glycosome are peroxins (PEXs) 5 and 7 [56]. Remarkably, both proteins, PEX5 (LINF_350019100) and PEX7 (LINF_290012400), were identified in the experimental proteome of *L. infantum* promastigotes. Moreover, we made a direct comparison between the *L. infantum* proteome reported here and two studies focused on glycosomal proteomes in *Leishmania tarentolae* and *L. donovani* [48,57]. Colasante and collaborators [48] identified 464 proteins in a glycosomal membrane preparation from *L. tarentolae*, and they concluded that 258 would be glycosomal proteins, including 40 enzymes. Interestingly, the orthologs of 165 (64%) of these proteins were experimentally identified in the *L. infantum* promastigote proteome. In particular, complete enzymatic complements involved in glycosomal glycolysis and gluconeogenesis steps were identified in both studies. Jamdhade and coworkers [57] reported the proteome analysis of an enriched glycosome fraction from *L. donovani* promastigotes, identifying 1355 proteins. In our study, orthologs to 853 of those putative *L. donovani* glycosomal proteins were found; these are listed in Appendix A.

The purine salvage pathway, essential for trypanosomes, also takes place in glycosomes [58]. Notably, the 13 enzymes composing the purine salvage pathway (Figure 4) [58] were identified in this *L. infantum* experimental proteome, which was in agreement with the relevance of this metabolic route for parasite survival. In this regard, it is somewhat unexpected that only two enzymes (adenylosuccinate synthetase (ADSS) and inosine monophosphate dehydrogenase (IMPDH)) were identified in the *L. tarentolae* glycosomal proteome, and five of these enzymes were identified in the glycosomal fractions of *L. donovani*. Similarly, we identified most of the enzymes constituting the de novo pyrimidine biosynthesis pathway (Table 4) [59] in the *L. infantum* proteome, whereas Jamdhade et al. [57] only found one enzyme of this pathway in the *L. donovani* glycosomal proteome—the orotate phosphoribosyltransferase (LDBPK_160560).

### 3.6. Exoproteome Components Identified in the L. infantum Experimental Proteome

*Leishmania*-secreted molecules and exosomes are particularly relevant for infection establishment because parasites and exosomes are co-egested during the insect blood meal [60]. A detailed analysis of the *L. infantum* secreted proteins (exoproteome) was carried out by Santarem et al. [35]. These authors found that the proteome profiles were distinct depending on the metabolic stage of the parasites (logarithmic or stationary phase promastigotes). The number of distinct proteins identified in that study was 297, and around 90% of them were also identified in the proteome reported here. In another outstanding study, Atayde et al. [61] analyzed the proteomic composition of *L. infantum* exosomes and extracellular vesicles that were directly isolated from the sand fly midgut. Table 5 lists proteins commonly present in exosome preparations; all of them were identified in the *L. infantum* experimental proteome reported here.

### 3.7. Other Relevant Proteins Identified in the L. infantum Promastigote Proteome

Proteins with a high molecular weight (HMW) represent a challenge for mass spectrometry-based assays, as they are usually underrepresented in protein extracts used for proteomic analysis. To overcome this limitation, Brotherton et al. [63] optimized extraction protocols to enrich HMW proteins and membrane proteins in *L. infantum* promastigotes and amastigotes. In our study, we confirmed the presence of tryptic and/or chymotryptic peptides from 35 HMW proteins with a molecular weight (MW) higher than 200 kDa (Appendix A). Among them, the identification of a calpain-like cysteine peptidase was remarkable, as it had an estimated MW of around 700 kDa (LINF_270010200) and was identified by 124 unique peptides, thus covering 24% of the amino acid sequence.

The flagellum is a characteristic organelle of *Leishmania* that confers motility to the parasite in the promastigote stage, during which this structure is particularly prominent. In a recent publication, an exhaustive structural and functional characterization of the *L. mexicana* promastigote flagellum was reported [64]. In that study, flagella preparations were analyzed by proteomics, and this allowed for the identification of 701 unique proteins for this organelle. Orthologues to around 400 flagellum-specific proteins were identified in the *L. infantum* proteome described here. More importantly, most of the proteins relevant for flagellum assembly and motility in *L. mexicana* promastigotes [64] were identified in the experimental proteome of *L. infantum* promastigotes (Appendix A).

### 3.8. Detection of Post-Translational Modifications

The PTMs of proteins influence their activity, structure, turnover, localization, and capacity to interact with other proteins. In *Leishmania*, PTMs, together with mRNA stability and translation processes, are the essential regulators of gene expression. In this study, based on MS/MS spectra, a significant number of phosphorylated, methylated, acetylated, glycosylated, and/or formylated proteins were identified in the *L. infantum* proteome. Thus, even though specific enrichments of modified proteins were not performed, we identified modified peptides that accounted for 10 phosphorylated, 144 methylated, 192 acetylated, 28 formylated, and 3 glycosylated proteins.

The phosphorylation of serine (S), threonine (T), and tyrosine (Y) amino acids implies an increase of 79.97 Da in their molecular weights (unimod.org). The phosphorylated proteins identified in this study and the modified residues are listed in Table 6. Apart from two unknown phosphoproteins (LINF_040005600 and LINF_220013200), the ribosomal protein S10, α tubulins, an rRNA biogenesis protein-like protein, the 3-ketoacyl CoA-thiolase, the flagellar protein KHARON1, the glycogen synthase kinase 3 (GSK-3), and the prototypical HSP70 might be regulated by phosphorylation (Table 6). In fact, the phosphorylation of HSP70 has been reported to occur during the stress response in both promastigotes and amastigotes of *L. donovani* [65].

Most of the observed phosphorylation events occurred on S and T, but in some proteins, phosphorylation on Y residues was also detected. Rosenzweig et al. [66] identified 16 phosphorylated proteins in *L. donovani* in either promastigotes or amastigotes; in that study, all of phosphorylations occurred at S or/and T residues. No coincidences exist between the phosphoproteins identified by these authors and those identified in this study (Table 6); however, this was not unexpected because phosphorylation is a dynamic modification and the numbers of phosphorylated proteins identified in both studies were low. Though the occurrence of phosphorylation events has been proven to be much lower in tyrosine residues [67], it is remarkable that phosphorylated tyrosines were found in α tubulins, the rRNA biogenesis protein, and in GSK-3 (Table 6). Some of these phosphorylated proteins have also been identified in previous studies. Thus, for instance, the 3-ketoacyl-CoA thiolase was found to be phosphorylated (at serine 229) in *L. donovani* promastigotes [68]. Kinases and phosphatases are enzymes implicated in the regulation of phosphorylation/dephosphorylation processes, and, accordingly, several serine, threonine, and tyrosine kinases and phosphatases have been identified in the *L. infantum* proteome (Appendix A).

Methylation (+14 Da) is a physiological PTM that occurs at the C- and N-terminal ends of proteins, and on the side chain nitrogen of arginine (R) and lysine (K); this modification is critical for regulating several cellular processes. Apart from those amino acids, methylations have been found to occur in other amino acids like aspartic acid (D), glutamic acid (E), histidine (H), glutamine (Q), and asparagine (N) [69]. In the *L. infantum* proteome, 139 proteins were predicted to be methylated, 76 of them showed methylation at K or R residues, 123 of the modified proteins showed D and/or E methylated residues, and a methylated-H was found in β-tubulin. All the methylated proteins detected in this study are listed in Appendix A. In summary, our findings pointed out that methylation at D and E residues would be relatively frequent in *Leishmania*; as suggested by Sprung et al. [69], methylations at E and D residues would increase the hydrophobicity of the modified proteins. Some examples of highly methylated proteins identified in this study are α and β tubulins, heat shock proteins HSP70 and HSP83/90, and the elongation factor 2 (eEF2). Many orthologs to the methylated proteins detected in this study were also identified as methylated in *L. donovani* [66].

Acetylation (+42.02 Da) is a PTM considered as relevant as phosphorylation in metabolic and signaling pathways. K acetylation has been described as a reversible enzymatic reaction that regulates protein function, as it is particularly relevant in chromatin compaction by the acetylation of histones [70]. Interestingly, the accumulation of acetylated histones has been observed at the polycistronic transcription initiation sites in *L. major* and *Trypanosoma cruzi* [71,72]. However, in the *L. infantum* proteomics data, peptides bearing acetylated K belonging to histones were not identified, as was the case in the *L. donovani* proteome [66], thus suggesting a relatively low proportion of acetylated histones in the bulk of total cellular histones. Some examples of proteins detected as acetylated in this study are β tubulins (LINF_330015100, LINF_330015200, and LINF_210028500; modified at K297), guanylate kinase (LINF_330018400, K3), the subunit β of ATP synthase (LINF_250018000; K511) and a calpain-like cysteine peptidase (LINF_140014400; K74).

In addition, acetylation at the N-terminal ends of proteins may occur either co- or post-translationally, as it is a frequent modification in eukaryotic proteins even though their physiological consequences remain poorly understood [73]. This irreversible modification affects protein fate in the cell and is carried out by N-terminal acetyltransferases (Nat). In the *L. infantum* promastigote proteome, we identified three of these enzymes: Nat-1, Nat-B, and Nat-C (Appendix A). On the other hand, among the 144 N-terminally acetylated proteins identified in this study (Appendix A), 40 proteins showed acetylation at their initial methionine (iM), and 104 were acetylated at the second amino acid, suggesting a cleavage of the iM during protein maturation [74]. In the cases in which acetylation takes place at the iM, we detected a bias in the amino acids located behind the iM. Thus, in 40% of the acetylated proteins, the second amino acid was the polar non-charged N or Q residues (Figure 5A also shows the other more frequent amino acids located at the second position). An acetylation reaction after iM removal was mainly found on S (55% of the cases) and alanine (A) (in 31%) residues (Figure 5B). These two amino acids, as well as threonine (found in 7.7% of the detected acetylated residues), have small side chains, a feature previously noted to favor a more efficient iM cleavage in the course of protein maturation [66,75].

On the other hand, the analysis of acetylated peptides allowed us to correct the initiator AUG codon (and, therefore, the predicted amino acid sequence) of four previously misannotated genes (whose new coordinates are indicated in Appendix A). One example is illustrated in Figure 5C; the coding sequence of the LINF_010008200 gene (coding for a poly (A) export protein) should be extended upstream 36 triplets based on the existence of an acetylated peptide encoded in that region.

Glycosylation also plays a relevant role in protein maturation, as well as in signal transduction mechanisms [76]. In this *L. infantum* proteomic study, we detected hexosylations or N-acetylhexosamine addition in three proteins; these modifications were consistent with N-linked glycosylation events at N residues. These PTM-modified proteins are cysteine peptidase A (CPA; LINF_190020600) modified at N345, 3-hydroxy-3 methylglutaryl CoA synthase (at N340; LINF_240027300), and PUF6 (at N483; LINF_330019100). The characterization of glycoproteins and the nature of their modifications remain challenging due to the complexity and variety of glycan moieties. In this study, two single modifications (hexosylation and N-acetylhexosamine addition) were searched, and this explained the extremely low number of detected glycosylated proteins. Rosenzweig et al. [66] found 13 glycosylated proteins—only one in asparagine and the rest in serine and threonine residues.

Formylation has not been previously described in trypanosomatids, but N-terminal methionine formylation in eukaryotes has been linked to cellular stress and protein degradation processes [77]. In particular, formyl-lysine residues has been found in histones and other nuclear proteins [78]. In our study, eight proteins were detected as formylated, mainly at leucine (in β-tubulin and typaredoxin), glycine (in a hypothetical protein; LINF_260024300), serine (in an RNA-guanylyltransferase) and lysine (in HSP83/90, a paraflagellar rod protein and a transaldolase) residues (see Appendix A). Future research on protein formylations in *Leishmania* and other organisms will provide insight into the physiological significance of this kind of PTM.

### 3.9. Proteogenomics

After assembling a genome, dedicated programs conduct the automatic annotation of ORFs. However, this annotation is not definitive, and a continuous effort of curation is needed. Transcriptomic analysis enables the obtainment of complete gene model annotation, including untranslated regions that are key to understand post-transcriptional regulation mechanisms. In addition, a proteogenomic analysis, such as that reported here, represents a powerful and useful approach for the identification of non-annotated genes, the correction of misannotations, or the validation of gene annotations [23]. For this purpose, in this work, the experimentally obtained peptide spectra were searched against all the polypeptides longer than 20 amino acids predicted from the ORFs found in the six possible translation frames of the recently re-sequenced genome for *L. infantum* JPCM5 strain [13]. The majority of the identified peptides fit well in current gene annotation (available at TriTryDB.org and http://leish-esp.cbm.uam.es/). Nevertheless, some peptides mapped to non-annotated coding-regions in the *L. infantum* (JPCM5) genome, uncovering eight novel protein-coding genes (Appendix A). Figure 6 shows an example of a novel hypothetical protein found in chromosome 27, together with the MS spectra of the two peptides that allowed for its identification.

As mentioned above (and illustrated in Figure 5), some of the detected peptides were mapped to regions located upstream of annotated coding sequences (CDS). This led to the addition of N-terminal extensions to 34 annotated proteins and the establishment of new translation start codons for their corresponding genes (Appendix A). All the detected peptides were confirmed to be unique, and the accuracy of their MS/MS spectra was manually revised.

## 4. Conclusions

In the last years, the characterization of trypanosomatid proteomes has become an active area of research. Here, we reported a proteomic analysis of *L. infantum*’s (JPCM5 strain) promastigote stage, and it was the first whole proteomic study in this species after the re-sequencing and de novo assembly of its genome in 2017 by González-De la Fuente et al. [13]. In addition, the search of the MS/MS spectra was performed against any possible ORFs larger than 20 triplets that existed in all-six frames from the *L. infantum* genome sequence. As a result, we identified 2352 proteins (Appendix A), most of them corresponding to the predicted sequences in current gene annotations (TriTrypDB.org). Comparisons between the results of this study and previous proteomics data derived from promastigote stages in different *Leishmania* species showed a significant level of similarity regarding the type of detected proteins. Nevertheless, this proteomic study showed experimental evidence on the expression in this parasite stage of 123 proteins that were not detected in previous studies; these proteins are listed in Appendix A. In addition, this study allowed for the identification of several PTMs in proteins, such as phosphorylation, methylation, acetylation, glycosylation, and formylation. Finally, this study also allowed for the identification of eight new protein-coding genes and the extension of the ORFs for 34. In conclusion, whole proteomics and genomic studies are inextricable, the results of the former depend on an accurate genome annotation, and a genome cannot be only annotated in an automatics manner. Thus, the proteomics data obtained in this study have allowed for the correction of annotation mistakes, the discovery of new genes, and experimental evidence of the existence of a large number of proteins that had to date been annotated as hypothetical.

All this new information, at the level of individual genes, is already available at Wikidata.org (searchable by the ID gene) and is going to be incorporated in the TriTrypDB database and the Leish-ESP website (http://leish-esp.cbm.uam.es/).

## Figures and Tables

**Figure 1 genes-11-01036-f001:**
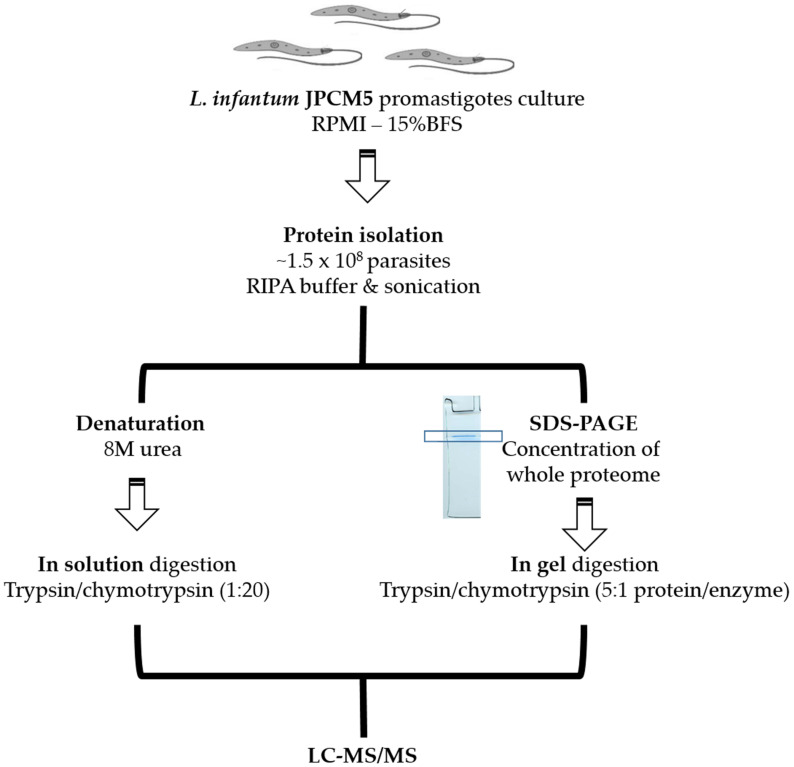
Workflow for protein extraction and proteomic analyses of *Leishmania infantum* promastigotes. The experimental MS data were searched against the UniProt protein database and a database consisting of all possible polypeptides encoded in the six-frames of the *L. infantum* genome (based on v2/2018; www.leish-esp.cbm.uam.es; [13]).

**Figure 2 genes-11-01036-f002:**
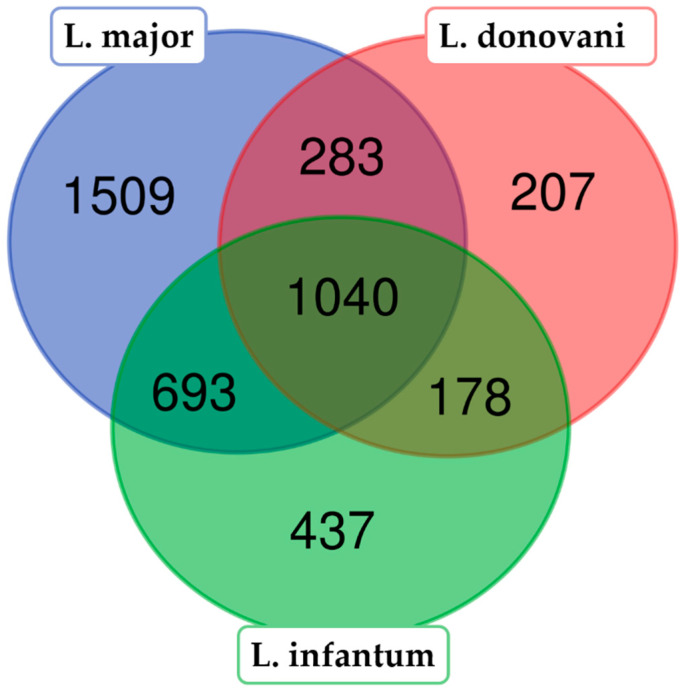
Comparison (Venn diagram) between the identified proteins in this work (*L. infantum*, in green) and those identified in two previous studies [20,36] performing proteomic analysis in *Leishmania donovani* (in red) and *Leishmania major* (in blue). The Venn diagram was created by the tool available at bioinformatics.psb.ugent.be/webtools/. Note—the discrepancy between the number of proteins identified by Rosenzweig et al. [20] (see text) and those represented in the Venn diagram (1713 vs. 1708) was due to 5 gene duplications that were corrected after re-assembling of the *L. infantum* genome [13].

**Figure 3 genes-11-01036-f003:**
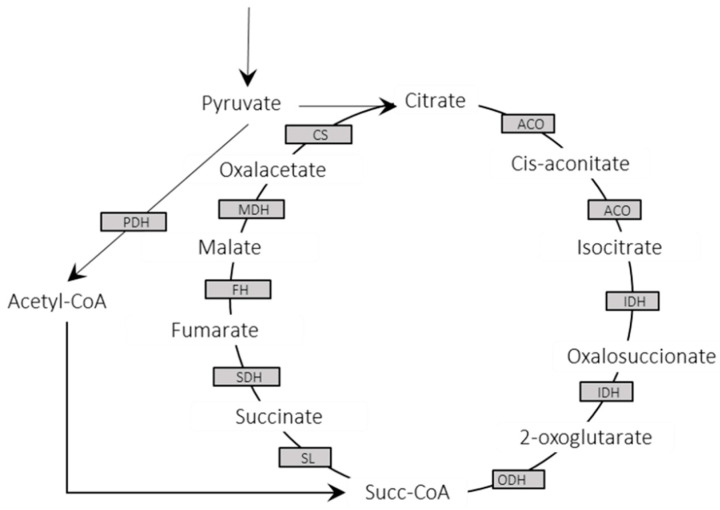
Detected enzymes in the *L. infantum* JPCM5 proteome composing the complete tricarboxylic acid (TCA) cycle. PDH: pyruvate dehydrogenase; ACO: aconitase; IDH: isocitrate dehydrogenase; ODH: 2-oxoglutarate dehydrogenase; SL: succinyl-CoA ligase; SDH: succinyl dehydrogenase; FH: fumarate hydratase; MDH: malate dehydrogenase; and CS: citrate synthase.

**Figure 4 genes-11-01036-f004:**
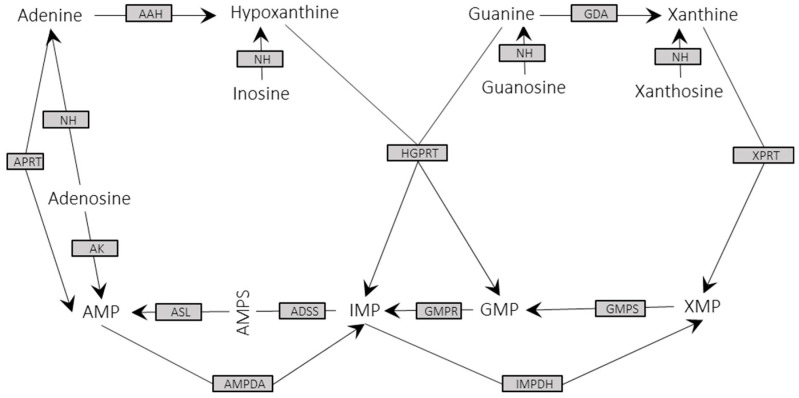
Enzymes from the purine salvage cycles identified in the *L. infantum* experimental proteome. All the enzymes (grey squares) that are required to complete the pathway were identified in this study. APRT: adenine phosphoribosyltransferase (LINF_130016900); NH: nucleoside hydrolase (LINF_180021400); AK: adenosine kinase (LINF_300014400); AAH: adenine aminohydrolase (LINF 350026800); ASL: adenylsuccinate lyase (LINF 040009600); AMPDA: AMP deaminase (LINF 130014700); ADSS: adenylosuccinate synthetase (LINF 130016900); GMPR: GMP reductase (LINF 170014800); GMPS: GMP synthase (LINF 220006100); HGPRT: hypoxanthine-guanine phosphoribosyltransferase (LINF 210014900); GDA: guanine deaminase (LINF 290014000); IMPDH: inosine monophosphate dehydrogenase (LINF 190022000); and XPRT: xanthine phosphoribosyltransferase (LINF 210015000). The cycle was depicted according to Boitz et al. [58].

**Figure 5 genes-11-01036-f005:**
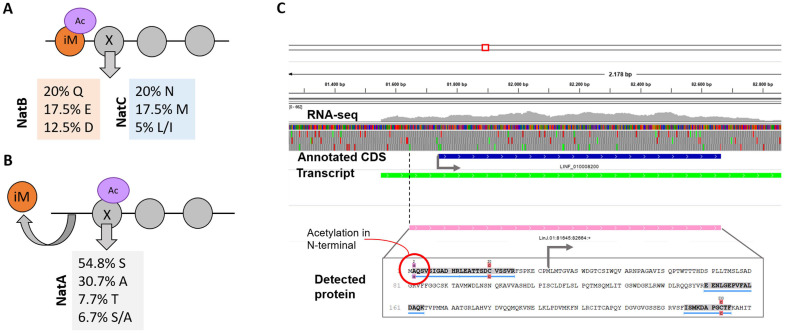
Features of the N-terminal acetylated proteins identified in this study. (**A**) Frequencies (percentages) of amino acids found next to the acetylated initial methionine (iM) and the putative enzymes responsible for the acetylation. (**B**) Percentages of amino acids found to be acetylated after cleavage of the iM and the putative enzyme involved in the reaction. (**C**) An example illustrating the usefulness of proteomics data for improving gene annotations. An acetylated peptide (red circle on the grey shaded sequence) was mapped to sequences located upstream of the currently annotated coding sequence for gene LINF_010008200 (blue box). The corrected gene (pink box) fit well into the transcript (green box). The image in (**C**) was generated using the Integrative Genome Viewer (IGV). CDS: coding sequence.

**Figure 6 genes-11-01036-f006:**
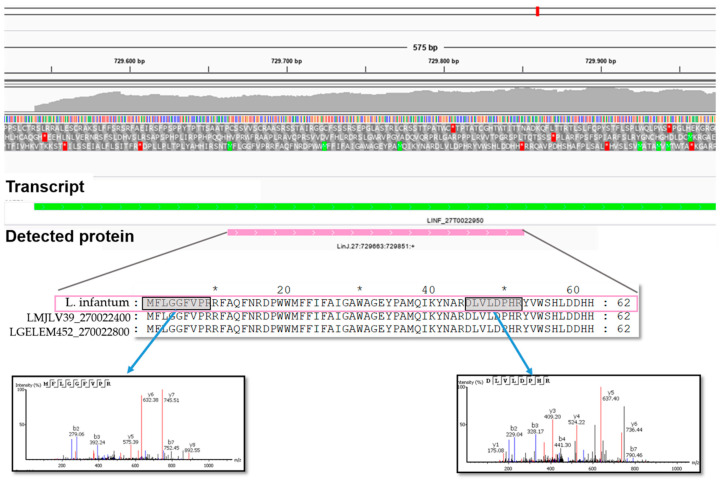
Identification of a novel protein based on the mapping of two experimentally detected-peptides in a region of *L. infantum* JPCM5 chromosome 27, which currently lacks an annotated ORF. The new CDS, named LINF_ 270,022,950 (pink box), fit well within a predicted transcript (LINF_27T0022950; green box). Interestingly, the predicted amino acid sequence was well-conserved when compared with proteins annotated in the genomic assemblies of *L. major* LV39 (ID: LmjVL39_270022400) and *Leishmania gerbilli* LEM452 (ID: LGELEM452_270022800).

**Table 1 genes-11-01036-t001:** List of glycosomal enzymes related to gluconeogenesis and glycolysis identified in *L. infantum* (according to the Kyoto Encyclopedia of Genes and Genomes (KEGG) pathway database).

Gene ID	Unique Peptides	Description
LINF_040016700	12	Fructose-1-6-bisphosphatase
LINF_120010600	43	Glucose-6-phosphate isomerase
LINF_200006000	41	Phosphoglycerate kinase C-glycosomal
LINF_210007800	47	Hexokinase
LINF_210012000	13	Phosphoglucomutase
LINF_230009500	10	Aldose 1-epimerase-like protein
LINF_240013700	32	Triosephosphate isomerase
LINF_250017300	46	Aldehyde dehydrogenase—mitochondrial precursor
LINF_270024900	56	Glycosomal phosphoenolpyruvate carboxykinase
LINF_290032900	23	ATP-dependent phosphofructokinase
LINF_300035000	49	Glyceraldehyde 3-phosphate dehydrogenase—glycosomal
LINF_300039500	13	PAS-domain containing phosphoglycerate kinase
LINF_340040800	13	Aldose 1-epimerase-like protein

**Table 2 genes-11-01036-t002:** List of cytosolic enzymes related to gluconeogenesis and glycolysis identified in the *L. infantum* proteome (according to the KEGG pathway database).

Gene ID	Unique Peptides	Description
LINF_140018000	55	Enolase
LINF_180019200	23	Pyruvate dehydrogenase e1 component α subunit
LINF_210011100	10	Dihydrolipoamide acetyltransferase
LINF_210012000	13	Phosphoglucomutase
LINF_230009000	43	NADP-dependent alcohol dehydrogenase
LINF_230014800	47	Acetyl-CoA synthetase
LINF_250023800	29	Pyruvate dehydrogenase e1 β subunit
LINF_290025700	4	Dihydrolipoamide dehydrogenase
LINF_310034500	2	Dihydrolipoamide dehydrogenase
LINF_320040600	37	Dihydrolipoamide dehydrogenase
LINF_350005300 LINF_350005400	43	Pyruvate kinase
LINF_360030600	43	Glyceraldehyde 3-phosphate dehydrogenase—cytosolic
LINF_360034400	19	Dihydrolipoamide acetyltransferase precursor

**Table 3 genes-11-01036-t003:** Identified molecular chaperones of *L. infantum* promastigotes, with putative mitochondrial location (according to Tasbihi et al. [22]).

Gene ID	Unique Peptides	Molecular Chaperone
LINF_260011400	9	Heat shock protein 10 (HSP10)
LINF_320040500	5	HSP40/JDP45
LINF_350035100	7	HSP40/JDP50
LINF_360027100	19	HSP60/cpn60.2
LINF_360027200	26	HSP60/cpn60.3
LINF_240010000	5	HSP40/JDP8
LINF_260017400	42	HSP70.4
LINF_280017800	51	Grp78/BiP
LINF_330033000	48	HSP75/TRAP-1
LINF_020012400	23	HSP78
LINF_330009000	22	HSP83/90

**Table 4 genes-11-01036-t004:** Enzymes involved in de novo pyrimidine biosynthesis identified in *L. infantum* promastigote proteome.

Gene ID	Unique Peptides	Description
LINF_060011200	7	Deoxyuridine triphosphatase
LINF_160010400	5	Dihydroorotate dehydrogenase (fumarate)
LINF_160010500	15	Aspartate carbamoyltransferase
LINF_160010700	28	Orotate phosphoribosyltransferase
LINF_160011200	6	Carbamoyl-phosphate synthase
LINF_180021400	17	Nonspecific nucleoside hydrolase
LINF_340016700	8	Uracil phosphoribosyltransferase

**Table 5 genes-11-01036-t005:** Common components of *Leishmania* exosomes identified in this *L. infantum* proteome study.

Gene ID	Description	Features [Ref.]
LINF 050017500; LINF 040007000	Surface antigens	Virulence factor [61]
LINF_090013900	Oligopeptidase b	Virulence factor [35]
LINF_100010100	GP63-leishmanolysin	Virulence factor [60,61]
LINF_120014700	Surface antigen protein 2	Virulence factor [61]
LINF_140018000	Enolase	Virulence factor [35,62]
LINF_150019000	Tryparedoxin peroxidase	Virulence factor [61]
LINF_170005900	Elongation factor 1-α	Exosome marker [61]
LINF_190020600	Cysteine peptidase A (CPA)	Virulence factor [61]
LINF_200018000	Calpain-like cysteine peptidase	Virulence factor [61]
LINF_280035000 LINF_280036000	HSP70	Exosome marker [61]
LINF_280034700	Receptor for activated C kinase 1	Immunomodulator [35]
LINF_320036700	Nucleoside diphosphate kinase b	Immunomodulator [35]
LINF_330009000	HSP83/90	Exosome marker [61]
LINF_350027300 LINF_350027500	Kinetoplastid membrane protein 11 (KMP11)	Immunomodulator [35]
LINF_360018400	Fructose-1-6-bisphosphate aldolase	Immunomodulator [60]

**Table 6 genes-11-01036-t006:** Phosphoproteins in the *L. infantum* proteome.

Gene ID	Description	Position
LINF_040005600	Hypothetical protein-conserved	T187
LINF_130007700; LINF_130007800; LINF_130008000; LINF_130008200; LINF_130008300; LINF_130008400; LINF_130008600; LINF_130008700	α tubulin	T334; Y357
LINF_180007700	Glycogen synthase kinase 3 (GSK-3)	Y186
LINF_190017000	Hypothetical protein-conserved	S120
LINF_200011800	rRNA biogenesis protein-like protein	Y571
LINF_220013200	Hypothetical protein-conserved	S45
LINF_230014600	3-ketoacyl-CoA thiolase	S229
LINF_280035400	HSP70	T159; T164
LINF_360015600; LINF_360015700	40S ribosomal protein S10	S157
LINF_360068400	Flagellum targeting protein KHARON1	S158

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
