# Peer review of "The Experimental Proteome of Leishmania infantum Promastigote and Its Usefulness for Improving Gene Annotations"

_genes, 2020, doi:10.3390/genes11091036_

Round 1

Reviewer 1 Report

The manuscript deals with the proteome of Leishmania infant a trypanosomatid responsible of visceral leishmaniasis in a high number of countries worldwide.The work presented at the manuscript improves previous ones as well as includes new data about protein modification sites probably involved in key functions of this parasite with important developmental features along its biological cycle.

Reviewer 2 Report

The article entitled “The experimental proteome of Leishmania infantum promastigote and its usefulness for improving gene annotations” authored by Sanchiz and collaborators, reports a proteomic analysis of L. infantum. This time, the authors set out to produce the proteome of one single developmental stage of axenic parasites. The absence of an experimental comparative group makes the article extremely descriptive and little stimulating reading.  The comparisons present in the article were made using other studies that were carried out employing different technologies and conditions. 

In my opinion the results of this new experimental L. infantum proteome don’t should be published in this form. It should be used as an element of comparison with another proteome produced and analyzed under the same conditions. For example, axenic amastigotes of L. infantum.  A comparative abordage with the pathogen in a different development stage could reveal new relevant aspects of parasite development.

Other observations:

Gender and species names must be written in italics. In the text, the names of the Leishmania species mentioned in the study are written in regular type.  Please fix It.

Fig 2.- The Venn Diagram

In figure 2, the authors compare their results with those of other studies. The red circle represents the results of Rosenzweig et al. (2008), where 1713 ptns were identified, but when we add the values ​​present in that circle, we only obtain 1708. How do the authors explain this divergence?

Finally, how do the authors explain the low number of glycosylated proteins, only 3 and that the abundant glycoprotein 63 (GP63) is not among them?
